# A New Shapley-Based Feature Selection Method in a Clinical Decision Support System for the Identification of Lung Diseases

**DOI:** 10.3390/diagnostics13233558

**Published:** 2023-11-28

**Authors:** Fevzi Yasin Kababulut, Damla Gürkan Kuntalp, Okan Düzyel, Nermin Özcan, Mehmet Kuntalp

**Affiliations:** 1Department of Electrical Electronic Engineering, Dokuz Eylül University, Izmir 35390, Turkey; damla.kuntalp@deu.edu.tr (D.G.K.); mehmet.kuntalp@deu.edu.tr (M.K.); 2Department of Electrical Electronic Engineering, Izmir Institute of Technology, Izmir 35433, Turkey; okanduzyel@iyte.edu.tr; 3Department of Biomedical Engineering, Iskenderun Technical University, Iskenderun 31200, Turkey; nermin.ozcan@iste.edu.tr

**Keywords:** decision tree, Shapley value, lung diseases, audio classification

## Abstract

The aim of this study is to propose a new feature selection method based on the class-based contribution of Shapley values. For this purpose, a clinical decision support system was developed to assist doctors in their diagnosis of lung diseases from lung sounds. The developed systems, which are based on the Decision Tree Algorithm (DTA), create a classification for five different cases: healthy and disease (URTI, COPD, Pneumonia, and Bronchiolitis) states. The most important reason for using a Decision Tree Classifier instead of other high-performance classifiers such as CNN and RNN is that the class contributions of Shapley values can be seen with this classifier. The systems developed consist of either a single DTA classifier or five parallel DTA classifiers each of which is optimized to make a binary classification such as healthy vs. others, COPD vs. Others, etc. Feature sets based on Power Spectral Density (PSD), Mel Frequency Cepstral Coefficients (MFCC), and statistical characteristics extracted from lung sound recordings were used in these classifications. The results indicate that employing features selected based on the class-based contribution of Shapley values, along with utilizing an ensemble (parallel) system, leads to improved classification performance compared to performances using either raw features alone or traditional use of Shapley values.

## 1. Introduction

Accurate diagnosis and classification of lung diseases are crucial for proper management and treatment. The ongoing COVID-19 pandemic has added to its importance, causing both short and long-term damage to the lungs [1,2,3]. In recent years, there has been a notable rise in studies focused on understanding the pathophysiology of lung diseases, resulting in the development of diverse diagnostic and classification tools [3,4,5]. A pulmonologist diagnoses a patient’s condition through anamnesis (complaints), physical examination, and, crucially, by listening to respiratory sounds (auscultation) [6]. Auscultation is particularly significant, and a diagnosis supported by accurately interpreting respiratory sounds, when complemented by other findings, has minimal margin for error. However, auscultation has the drawback of being subjective, relying on the doctor’s experience and external noise factors. To mitigate this issue, the classification of respiratory sounds recorded with a digital stethoscope through software can automatically detect the patient’s condition, reducing dependence on subjective interpretation [7].

Software for classifying respiratory sounds functions as an automatic classifier, incorporating machine learning algorithms. Examples of these algorithms include K-NN classifiers [8], Support Vector Machines [9], Decision Trees [10], and Ensemble Algorithms. These algorithms assign class labels to the data based on knowledge derived from training data. In the literature, certain studies concentrate on identifying wheeze and crackle sounds in pulmonary records, often associated with unhealthy cases [11,12,13,14,15,16]. Other studies explore the classification of prevalent lung diseases such as pneumonia, asthma, and COPD [16,17,18,19,20,21,22,23]. Below, we provide an overview of both types of studies.

Numerous studies have explored the correlation between lung sounds and various lung diseases, resulting in the creation of diverse classification systems based on auscultation findings. For instance, Kim Y., Hyon Y., Jung S.S., Lee S., Yoo G., Chung C., and Ha T. reported a higher prevalence of wheezes in patients with asthma and COPD, while crackles were more common in patients with pneumonia, interstitial pulmonary fibrosis (IPF), and pulmonary edema [24]. Moran-Mendoza O., Ritchie T., and Aldhaheri S. found an association between fine and coarse crackles and idiopathic pulmonary fibrosis (IPF) [25]. In their study, they classified seven different audio-based classes (normal, roughness, coarse crackling, monophonic wheeze, polyphonic wheeze, stridor, and squawk) from lung sound signals (LSS) using Random Forest (RF), Adaboost, and Gradient Boosting (GB) algorithms [26]. Lung sound signals (LSS) underwent Discrete Wavelet Transform (DWT) using a fourth-order Daubechies main wavelet (db4). Additionally, 12 Mel Frequency Cepstral Coefficients (MFCC) features were extracted with a 60% overlap of LSS. Using the db4 features, Random Forest (RF), Adaboost, and Gradient Boosting (GB) algorithms achieved accuracies of 99.04%, 96.63%, and 95.11%, respectively. With MFCC features, the accuracies were 98.76%, 96.29%, and 94.71% for RF, Adaboost, and GB algorithms, respectively.

Several studies, including ours, focus on classifying lung sound recordings from the ICBHI database. Notably, many of these studies [20,27,28,29,30] employ the CNN algorithm, while others explore alternative approaches such as MLP Artificial Neural Networks [31] and Boosted Decision Tree and SVM [21]. Diverse feature sets are utilized across these studies; for instance, some [20,28,29,30] leverage MFCC or Mel Spectrogram features, while one uses CWT [27], one employs LPCC [31], and another one adopts entropy-based features [21]. Classification targets also vary, with some studies [20,21,27,29] categorizing lung diseases into 5, 6, and 7 classes, while others [20,27] classify them into 3 classes: chronic, non-chronic, and healthy. Notably, one study [28] focuses on wheezing, crackling, and others, while another [30] aims to detect COPD. Another study [31] classifies lung sounds into healthy and unhealthy categories. Some studies [32] share our use of spectral information as features.

This study aims to enhance the performance of a lung disease classification system by incorporating Shapley values for feature analysis, which is a relatively new approach not widely explored in existing literature for lung disease diagnosis. Additionally, this study proposes a novel class contribution-based Shapley feature selection method, contributing to the advancement of methodologies in this domain. Although structures like CNN, ANN, and RNN are widely used, the DTA classifier was chosen in this study due to the fact that the class contributions of Shapley values can be seen with this type of classifier.

In this study, a 5-class lung disease classification was performed using the Decision Tree Algorithm (DTA) as the classifier with different feature sets. Initially, the classification was conducted using all extracted features, including Mel Frequency Cepstral Coefficients (MFCC), Power Spectral Density (PSD), and statistical features from lung sounds. This feature set, comprising 14 MFCC, 43 PSD, and 11 STF features, totaling 68 features, is denoted as Feature Set 1. The classification system using this feature set is referred to as the *first classification method (Method 1)*. Subsequently, the most efficient features, determined by their highest Shapley values for the 5-class classification problem, were ranked. A new reduced feature set (Feature Set 2) was then created using only these features. The same Decision Tree Algorithm (DTA) classifier was employed with this new feature set, constituting the *second classification method (Method 2)*. The next two methods involved an ensemble of five different DTAs, each dedicated to one class. In these methods, distinct feature sets were generated, considering the individual class contributions of the features. For the *third classification method (Method 3)*, this was accomplished by selecting features with the highest Shapley values specific to each separate class. These feature sets are denoted as Feature Set 3. In the *fourth classification method (Method 4)*, feature sets are created based on the results of 2-class lung disease classifications, such as ‘Healthy’ vs. ‘Others’ or ‘COPD’ vs. ‘Other’. For each binary classification, we obtained the ranking of the most effective feature subsets with the highest Shapley values. The best features were then selected for five different binary classifications separately, and these feature sets are referred to as Feature Set 4. In the last two methods (ensemble methods), five independent classification systems, each comprising a different Decision Tree Algorithm (DTA), were integrated into a single decision system. The final decision of the overall system depends on the independent decisions of each separate system.

In Section 2, we detail the database used. Section 3 covers the extracted features, while Section 4 presents the Decision Tree Algorithm (DTA) classifier and the parameters utilized. Section 5 explains the selection of feature subsets based on their Shapley values. Section 6 delves into the performance evaluation and improvements achieved with the use of Shapley values. The effects of feature selection on performance are discussed in Section 7. Finally, the Conclusion discusses the results and presents possible future works.

## 2. Database

The International Conference on Biomedical and Health Informatics (ICBHI) 2017 Lung Sounds Database, a public database created through the collaboration of researchers from two countries, is utilized in this study [33]. Comprising 920 lung sounds from 126 individuals, the database includes a total of 6898 respiratory cycles. The sample frequencies of the sounds in the datasets range from 4 kHz to 44.1 kHz, and the recordings vary in duration from 10 s to 90 s. Table 1 illustrates the distribution of the data across 8 classes, including ‘healthy’ and 7 frequently encountered lung diseases.

We excluded the records of Lower Respiratory Tract Infection (LRTI) and asthma due to the limited number of recordings for these labels. Out of the remaining lung sound recordings, 641 were used for training, and 276 for testing. To identify and, if necessary, remove low-quality recordings from the dataset, we calculated the Linear Predictive Cepstral Coefficient (LPCC) values. In the training data, LPCC values of 0 were found in 4 recordings in the ‘Healthy’ (1st) class, 8 recordings in the Upper Respiratory Tract Infection (URTI) (2nd) class, 46 recordings in the COPD (3rd) class, 3 recordings in the Bronchectasy (4th) class, and 1 recording in the Bronchiolitis (5th) class. In the test data, 4 recordings in the 1st class, 16 in the 3rd class, and 3 in the 4th class had an LPCC value of 0. Due to these recordings having very low Signal-to-Noise Ratio (SNR) values, they were removed from the dataset. Following this operation, the Bronchectasy class had very few remaining recordings and was consequently removed from the dataset, reducing the number of classes to 5 in the study. As depicted in Table 1, the data exhibits an unbalanced distribution, with 86% of records belonging to patients with COPD. To address this imbalance, data augmentation was implemented using information from the annotation files in the database, which includes details about the number of respiratory cycles in a sound recording and their start and end times. By treating the respiratory cycles of non-COPD records as separate recordings and saving them as new data, we enhanced the homogeneity of the database distribution. Additionally, five different datasets, as outlined in Table 2, were prepared for use in the ensemble methods described later.

## 3. Extracted Features

One of the crucial factors influencing classification success is the choice of features. In the realm of audio data classification, Mel Frequency Cepstral Coefficients (MFCC) have been widely preferred in the literature [11,12,17,26,28,30,34,35,36,37,38,39,40], along with Power Spectral Density (PSD)-based features [41,42]. MFCC-based features yield favorable results in audio signal analysis, as they emulate the nonlinear frequency selectivity of the human ear [43]. This process typically begins by segmenting the data. Subsequently, the Discrete Fourier Transform (DFT) of these segments is computed. Following this, the signals pass through triangular bandpass filters, and their coefficients in the Mel Scale are calculated. Ultimately, the Mel Frequency Cepstral Coefficients (MFCC) are computed by applying the Discrete Cosine Transform along with a logarithmic operation to these coefficients [44]. Through these calculations, a total of 14 MFCC features are extracted for each data.

Power Spectral Density (PSD) characterizes the energy distribution of the signal across the frequency range. Various methods can be employed for PSD estimation. In this study, we utilized the Welch method [44] to calculate the PSD estimation of lung sounds at different sampling frequencies. Additionally, we computed the average power within different frequency bands [45,46,47]. Consequently, a total of 43 PSD-based features were extracted.

Statistical features (STF) derived from the time domain representation of signals find application in various domains [48,49,50,51], including the detection and classification of lung sounds [52,53]. Within these features, distribution features like mean and standard deviation are employed to differentiate bimodal distributed sounds like a wheeze, while high-order statistical features such as skewness and kurtosis help distinguish non-periodic sounds like a crackle [45]. In this study, we extracted and utilized 11 statistical features, including mean, standard deviation, skewness, kurtosis, root mean square, mean absolute deviation, median absolute deviation, peak-to-peak difference, crest factor, shape factor, and impulse factor [17].

## 4. Decision Tree Classifier

Decision Trees are estimation algorithms with a tree-like structure, determining the final solution through branching decisions based on a series of criteria in classification or regression problems [17,35,54,55]. At the nodes of the tree, the features of the data are considered and ranked from the first node with the highest information gain to the last branch. In a classification problem, the leaves of the tree represent the decided classes for the test data. The algorithm begins by calculating the Gini impurities (*GI*) of the features, as shown in Equation (1).
(1)$$GI_{k}=1-\sum_{i=1}^{j}p_{i}{\left(T,x\right)}^{2},$$
where *p_i_*(*T,x*) is the probability of the class *i* of the feature *x* according to the test data *T*. The conditional Gini impurity of the feature for l classes is calculated as follows:(2)$$CG(T,x)=\sum_{k=1}^{l}p_{k}GI_{k},$$
where *p_k_* is the probability of the class *k* of the feature, *GI_k_* is the Gini impurity of the class *k* of the feature. According to the test data whose class is desired to be decided, the information gain (*IG*) of the feature is calculated as follows: (3)IG(T,x)=GI(T)−CG(T,x),

The feature with the highest information gain serves as the first node, and the calculations continue until the last branch, completing the Decision Tree. The Decision Tree Algorithm offers various optimization parameters to enhance accuracy and model performance. Key parameters include the *maximum depth*, which determines the maximum number of layers in the Decision Tree. Increasing the maximum depth enables the model to capture more complex data relationships but may lead to overfitting. Another crucial parameter is the *minimum sample split*, setting the minimum number of samples required to split a node. Adjusting this parameter helps prevent the model from splitting nodes with too few samples, mitigating overfitting. Similarly, the *minimum sample leaf* parameter establishes the minimum number of samples required at a leaf node. Increasing this parameter helps prevent creating leaves with too few samples, reducing overfitting. The *maximum features* parameter determines the maximum number of features that can be used in each split. By reducing the number of features, the model can avoid complexity and overfitting. The *criterion* parameter dictates the function used to measure the quality of a split, with the two most common criteria being Gini impurity and entropy. Adjusting these parameters allows for the optimization of the Decision Tree model to achieve the highest accuracy and performance on the given data.

## 5. Feature Sets and Classification Methods

The Shapley value, a tool originating from game theory that assesses the average contribution of players to a game, finds application in various fields, from economics to signal processing [56,57,58]. In this study, we employed Shapley values to gauge the contribution of features to classifier performance. Leveraging the advantages of these values in feature selection and, consequently, dimension reduction, we utilized Python’s Shapley library in our algorithms to calculate Shapley values, as outlined in Equation (4).
(4)$$S_{i}=\frac{1}{NP}\sum_{Coalitions\notin i}\frac{MC_{i}}{NC}$$
where *NP* is the number of players, *MC_i_* is the marginal contribution of the *i*th player to the coalitions, and *NC* is the number of coalitions excluding the *i*th player.

Initially, we conducted feature extraction by segregating audio recordings from different patients into training and testing sets. Subsequently, in the Python environment, we implemented a 5-class Decision Tree Algorithm using all raw features (i.e., without incorporating Shapley values) across all datasets. This classifier employed all extracted features, including MFCC, PSD, and statistical features, representing the first classification method (Method 1), as illustrated in Figure 1. Following this, we calculated the Shapley values of the features for the modeled algorithms. In the subsequent step, we separately sorted the top 30 features with the highest Shapley values for our methods (Methods 2–4), while the remaining 38 features were not evaluated due to their low Shapley values. The features with the highest Shapley values are incrementally added one by one, starting from three and ultimately merging to form new feature sets totaling 30. We presented these feature sets in various combinations as input data to the algorithms one by one, aiming to identify the most successful one based on all three performance criteria (Total Accuracy, MCC, and F1). The order of importance of features (for Feature Sets 2 and 3) according to the Shapley values is illustrated in Figure 2. In Figure 2, the class contribution of the features to the performance, based on the Shapley values, is depicted in the 5-class Decision Tree Algorithm using all features. In this figure, 1 represents “Healthy” (magenta), 2 represents “URTI” (brown), 3 represents “COPD” (purple), 4 represents “Pneumonia” (blue), and 5 represents “Bronchiolitis” (green). The Shapley value-based selection of feature sets commences with a 5-class lung disease classification using the Decision Tree Algorithm (DTA) as the classifier. Subsequently, the rankings of the most efficient features (those with the highest Shapley values) for this 5-class classification are determined, leading to the creation of a feature subset (Feature Set 2) comprising only these features. The same DTA classifier is employed with this new feature set as the second classification method (Method 2), as depicted in Figure 3.

For the third method (Method 3), features with the highest Shapley values specific to each separate class are selected based on the Shapley values in Figure 2. This process results in the creation of different feature sets (Feature Set 3) for each binary classifier, as depicted in Figure 4. In the fourth method (Method 4), feature sets are formed based on the results of 2-class lung disease classifications, such as “Healthy” vs. “Others,” “COPD” vs. “Other,” as illustrated in Figure 5. For each binary classification, the ranking of the most effective feature subsets (those with the highest Shapley values), referred to as Feature Set 4, is obtained. This approach involves selecting the best features for five different binary classifications, each performed separately. An example of the “URTI” vs. “Other” binary classification is presented in Figure 6.

## 6. Performance Evaluation

In this study, we assessed our system’s capability to detect the presence of any lung disease based on lung sounds, employing three different performance criteria. We utilized ‘Total Accuracy’ as the primary performance criterion. Considering the unbalanced nature of our dataset, we also employed additional performance criteria, namely F1 and Matthews Correlation Coefficient (MCC), for control purposes [59,60]. Notably, similar results were observed regardless of the chosen criterion.

We ran our algorithms in the Python environment, using the feature sets explained in Section 5. Following feature selection, we combine the train and test features into a single dataset and subsequently conduct cross validation. We employ cross validation only to calculate performance metrics, not for feature selection. After identifying the most effective features among the five datasets based on Shapley values, ensemble systems were designed wherein the decisions from each classifier were collectively evaluated. Conceptually, these five two-class classifiers can be likened to doctors specializing in five different diagnoses (healthy, URTI, COPD, Pneumonia, and Bronchiolitis). Subsequently, another expert makes a diagnostic decision by evaluating the diagnoses provided by these five experts. The ensemble of five different Decision Tree Algorithms (DTAs) was based on two methods (Methods 3 and 4), each tailored for one class. These methods involved creating different feature sets for each class, considering the individual class contributions of features. In these ensemble systems, five independent classification systems (DTAs) were integrated into a single decision system. As illustrated in Figure 4 and Figure 5, the final decision of the overall system is determined by considering the decisions of all independent binary classifiers.

## 7. Results

Table 3 presents the 5-fold cross-validation performance of binary classifiers using raw features, while Table 4 and Table 5 display the performance evaluation of algorithms (Methods 3 and 4) after discarding features with low Shapley values. The tables present the performance values obtained through 5-fold cross validation for five datasets with 2-class distribution, showcasing the different success criteria of the most effective features created based on Shapley value-based selection. Comparing the results in Table 3, Table 4 and Table 5, significant improvements in MCC, F1, and Total Accuracy criteria were consistently observed across all classifications with the use of Shapley value-based selected features. In Classifiers 1, 2, 4, and 5, the best results were achieved using Feature Set 4, while in Classifier 3, Feature Set 3 yielded the optimal outcomes across all criteria. When comparing the performance of the classifiers, the order of success based on the Total Accuracy criterion is as follows: Classifier 3 (COPD and The Others), Classifier 5 (Bronchiolitis and The Others), Classifier 4 (Pneumonia and The Others), Classifier 1 (Healthy and The Others), and Classifier 2 (URTI and The Others).

In our ensemble system (Method 4), it is crucial to address how ties are resolved. When evaluating the ensemble system on our 5-class test data, consider a scenario where the test data is healthy. If the 1st system decision is ‘Healthy,’ the 4th system decision is ‘Pneumonia,’ and the other classifiers’ decisions are ‘Others,’ the ensemble system’s joint decision would be both ‘Healthy’ and ‘Pneumonia.’ In such cases, we opt to trust the decision of the more confident expert, accepting it as correct. This approach yields a Total Accuracy value of 69.67%. Table 6 illustrates that the best results were achieved by Method 4 using Feature Set 4, which entails the ranking of the 30 features with the highest Shapley values in the 5 DTAs performing 2-class classification.

To compare the performances of Method 1 and Method 2, we examined the results of a single classifier that performs a 5-class classification. Table 6 illustrates that the best results for this single classifier were obtained by Method 2 using Feature Set 2, which consists of the ranking of the 30 features with the highest Shapley values in the 5-class Decision Tree Algorithm. Notably, compared to using all raw features (Feature Set 1), the utilization of a Shapley value-based reduced feature set (Feature Set 2) leads to a significant increase in Total Accuracy. Moreover, in the comparison of Shapley-based methods, the ensemble system (Method 4) using Feature Set 4 outperforms the 5-class Decision Tree Algorithm (Method 2) with Feature Set 2. This underscores the superior performance of the ensemble system over a single-classifier approach.

In the Decision Tree Algorithm (DTA), features are individually selected based on their information gain values, reflecting their contribution to classification. Conversely, features selected according to their Shapley values are ranked based on their share in the overall contribution to the algorithm’s performance when all features are used. Although the order of importance of features changes, we observed that common features were used in both methods. For Dataset 3, the 2nd and 18th features are common. The 18th feature is the 4th PSD feature. The 2nd feature corresponds to MFCC features, which are coefficients in the time domain with spectral information derived from 40 triangular filters. The center frequency of the first 13 filters is in the 0–1000 Hz frequency band, with a range of 133.33 Hz [61]. For the remaining 27 filters, the center frequency is 1.0711703 times higher than the frequency of the previous filter, covering the frequency band of 1000–6400 Hz [61]. Each MFCC coefficient represents log energy output in these 40 frequency bands due to its cosine transform. To determine the most effective frequency ranges of the MFCC features, you can identify the frequency band associated with filters having the highest coefficient log energy output in the cosine transform. The PSD feature operates in the frequency domain, and each feature corresponds to a specific range within the 0–22,500 Hz frequency band. Abnormal lung sounds, such as wheezing, have been reported in COPD patients [24], particularly in the frequency range of 100–2000 Hz [46]. In Dataset 3 of our study, the effective frequency range of the 2nd MFCC feature is 133.33–266.67 Hz, and the frequency range of the 4th PSD feature is 330.75–441.00 Hz, both falling within the specified frequency range in [46]. Similarly, in other datasets, we observed that the selected features were within the same frequency bands as the lung sounds specific to those diseases.

The detailed examination of Feature Set 4 used in Method 4, which yielded the most successful results, reveals interesting patterns. In Classifier 1 (H/O), 3 out of 7 features that provided the best results were MFCC, while 3 were PSD and 1 was STF. In Classifier 2 (U/O), 7 out of 12 features that provided the best results were MFCC, 4 were STF, and 1 was PSD. In the third classifier (C/O), 5 out of 7 features that provided the best results were MFCC, and 2 were PSD features. In Classifier 4 (P/O), 13 out of 28 features that provided the best results were MFCC, 9 were STF, and 6 were PSD features. For Classifier 5 (H/O), 11 out of 23 features that provided the best results were MFCC, 7 were STF, and 5 were PSD features. These results suggest that MFCC features contribute more to the classification results compared to other features.

## 8. Conclusions

The aim of this study is to propose a new feature selection method based on the class-based contribution of Shapley values. For this purpose, a clinical decision support system was developed to assist doctors in their diagnosis of lung diseases from lung sounds by determining whether people whose lung sound recordings were taken have a lung disease or not and, if yes, which diseases they have. These diseases include URTI, COPD, Pneumonia, and Bronchiolitis. We have not seen a study in the literature that uses Shapley values for feature selection in the diagnosis of lung diseases from lung sound recordings. Features based on Power Spectral Density (PSD), Mel Frequency Cepstral Coefficients (MFCC), and statistical characteristics extracted from lung sound recordings were used in these classifications. We generated various feature sets using methods based on Shapley values including a new one proposed by us that evaluates Shapley values on a class basis.

All of the developed systems use the Decision Tree Algorithm (DTA) classifier. The systems developed consist of either a single DTA classifier or five parallel DTA classifiers, each of which is optimized to make a binary classification, such as healthy vs. others, COPD vs. Others, etc. The most important reason for using a Decision Tree Classifier instead of other high-performance classifiers such as CNN and RNN is that the class contributions of Shapley values can be seen with this classifier.

A distinction of our study from similar ones in the literature is that instead of extracting image-based features, we used 3 different types of features (MFCC, PSD, and STF) obtained from raw 1D lung sounds. We were able to find a study in the literature that classifies with 1D features like ours. In [34], MFCC features were extracted for different window sizes and window steps of lung sounds in the ICBHI dataset. In this study, 99% sensitivity performance was obtained in healthy–unhealthy classification with the LSTM algorithm, while in our study, 71.50% sensitivity performance was obtained with the Decision Tree Algorithm.

The results indicate that employing features selected based on the class-based contribution of Shapley values, along with utilizing an ensemble (parallel) system, leads to improved classification performance compared to performances using either raw features alone or traditional use of Shapley values. While our primary focus is on comparing the use of our proposed method with the normal use of Shapley values rather than designing a system with superior accuracy, we also conducted a comparison with another similar ensemble system from the literature, which, although it did not employ Shapley values, utilized the dataset we used. While the ensemble system in this study [5] classified abnormal and normal lung sounds with 66.70% accuracy, our ensemble system (Method 4) achieved a more successful result with 69.67% accuracy in a 5-class classification. It is also worth noting that our algorithms are independent of the type of stethoscope and the regions where the lung sounds are listened from. Thus, unlike some studies in the literature, we have designed algorithms that are independent of these parameters, sacrificing the high accuracy value in our study.

In addition, the Decision Tree Algorithm allows us to identify important features in disease diagnosis by examining the tree structure. In our analysis, we have seen that the frequency bands of the important MFCC and PSD features are the same as the frequency bands of the distinctive sounds of lung diseases. In this way, we have also conducted a kind of validation of our system.

The International Conference on Biomedical and Health Informatics (ICBHI) 2017 Lung Sounds Database, which was created with the contributions of researchers from two countries, is used in this study. In this dataset, the ratio of records in the COPD class to all records is 45.91%, while the ratio of records belonging to the other 4 classes to all records is 54.09%. This highly unbalanced dataset inevitably causes the performance of classifiers to be poorer. As a future work of this study, we are planning to eliminate the negativity created by the unbalanced data distribution between different diseases by data augmentation techniques. For this purpose, we will try to generate new synthetic disease data for smaller classes by using GAN architecture. In addition, we are planning to make a detailed comparison of our proposed method with other commonly used feature selection methods.

## Figures and Tables

**Figure 1 diagnostics-13-03558-f001:**
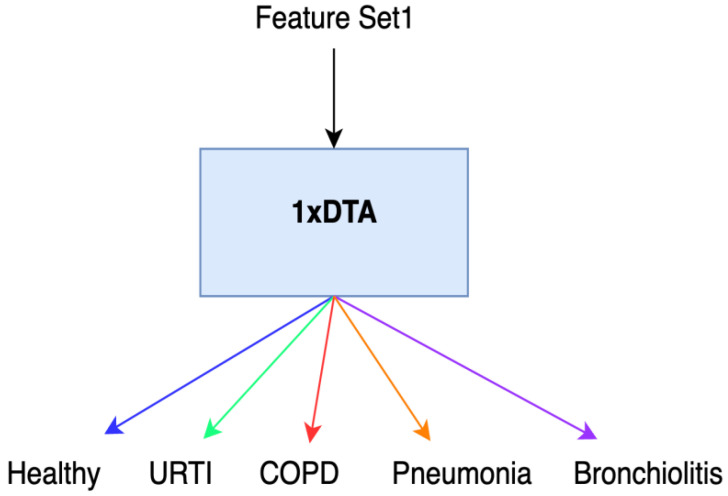
Method 1.

**Figure 2 diagnostics-13-03558-f002:**
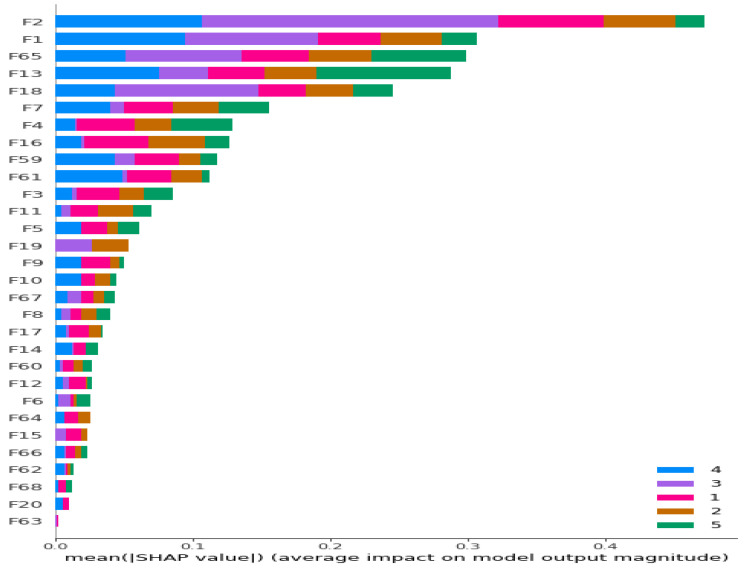
Ranking of the most contributing 30 features according to their Shapley values for 5-class classification problem. Different colors represent contributions of features to corresponding classes.

**Figure 3 diagnostics-13-03558-f003:**
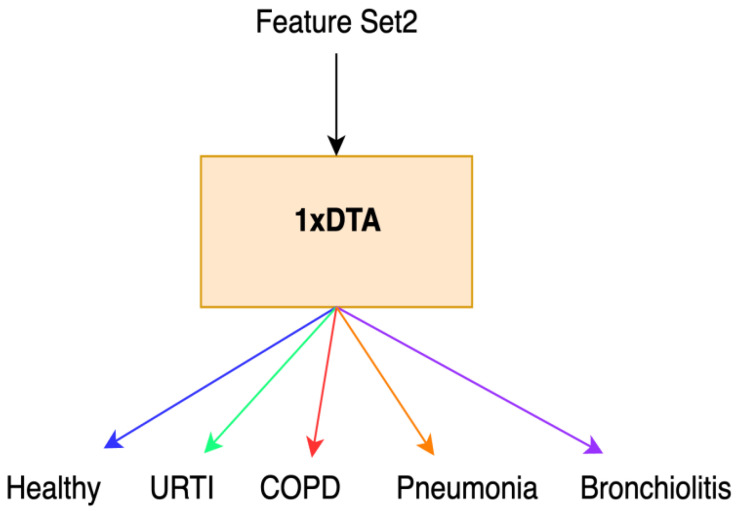
Method 2.

**Figure 4 diagnostics-13-03558-f004:**
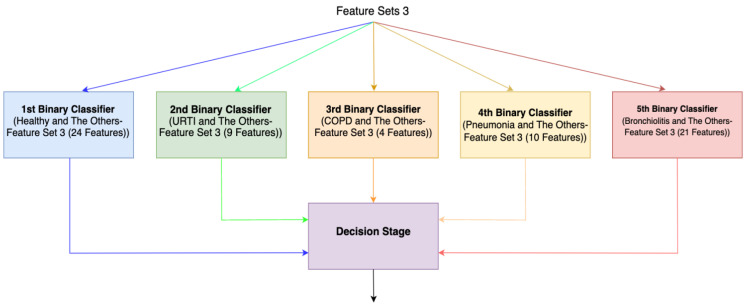
Ensemble system (Method 3).

**Figure 5 diagnostics-13-03558-f005:**
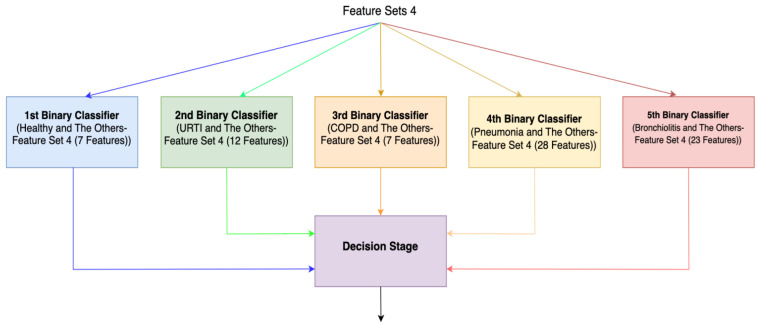
Ensemble system (Method 4).

**Figure 6 diagnostics-13-03558-f006:**
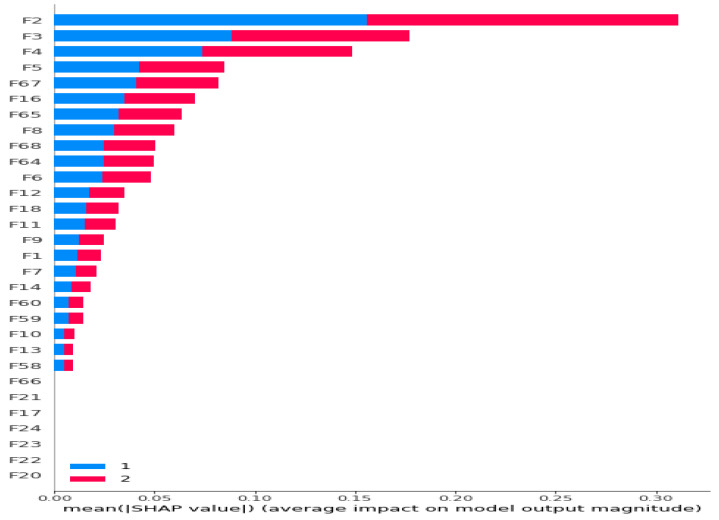
Ranking of all 68 features according to their Shapley values for Dataset 1 (URTI and The Others) (Bar Plot).

**Table 1 diagnostics-13-03558-t001:** Distribution of sound recordings of ICBHI 2017 database (in number of recordings).

Healthy	Asthma	URTI	COPD	LRTI	Bronchiectasis	Pneumonia	Bronchiolitis
35	1	23	793	2	16	37	13
(3.80%)	(0.11%)	(2.50%)	(86.20%)	(0.22%)	(1.74%)	(4.02%)	(1.41%)

**Table 2 diagnostics-13-03558-t002:** Distribution of datasets (in number of recordings).

**Dataset 1**			
**Healthy**		**Others**	
**Train**	**Test**	**Train**	**Test**
177	72	903	440
(11.12%)	(4.52%)	(56.72%)	(27.64%)
**Dataset 2**			
**URTI**		**Others**	
**Train**	**Test**	**Train**	**Test**
86	97	994	415
(5.40%)	(6.09%)	(62.44%)	(26.07%)
**Dataset 3**			
**COPD**		**Others**	
**Train**	**Test**	**Train**	**Test**
509	222	571	290
(31.97%)	(13.94%)	(35.87%)	(18.22%)
**Dataset 4**			
**Pneumonia**		**Others**	
**Train**	**Test**	**Train**	**Test**
204	81	876	431
(12.81%)	(5.09%)	(55.03%)	(27.07%)
**Dataset 5**			
**Bronchiolitis**		**Others**	
**Train**	**Test**	**Train**	**Test**
104	40	976	472
(6.53%)	(2.51%)	(61.31%)	(29.65%)

**Table 3 diagnostics-13-03558-t003:** The 5-fold cross-validation performance of 2-class classification using raw features.

Classifier/Criterion	1st Binary Classifier (H/O)	2nd Binary Classifier (U/O)	3rd Binary Classifier (C/O)	4th Binary Classifier (P/O)	5th Binary Classifier (B/O)
**Total Accuracy**	0.8234	0.8040	**0.9768**	0.8285	0.8028
**MCC**	0.3623	0.1541	**0.9540**	0.4616	0.3189
**F1**	0.6702	0.5526	**0.9767**	0.7125	0.6221

**Table 4 diagnostics-13-03558-t004:** The 5-fold cross-validation performance of 2-class classification of Method 3.

Classifier/Criterion	1st Binary Classifier (H/O)	2nd Binary Classifier (U/O)	3rd Binary Classifier (C/O)	4th Binary Classifier (P/O)	5th Binary Classifier (B/O)
**Total Accuracy**	0.8203	0.8178	**0.9931**	0.8260	0.8643
**MCC**	0.3832	0.2342	**0.9862**	0.4696	0.2524
**F1**	0.6770	0.5999	**0.9931**	0.7139	0.6041

**Table 5 diagnostics-13-03558-t005:** The 5-fold cross-validation performance of 2-class classification of Method 4.

Classifier/Criterion	1st Binary Classifier (H/O)	2nd Binary Classifier (U/O)	3rd Binary Classifier (C/O)	4th Binary Classifier (P/O)	5th Binary Classifier (B/O)
**Total Accuracy**	0.8442	0.8241	**0.9843**	0.8524	0.8851
**MCC**	0.4487	0.2497	**0.9686**	0.5557	0.3199
**F1**	0.7128	0.6057	**0.9842**	0.7568	0.6417

**Table 6 diagnostics-13-03558-t006:** Comparison of the performances of 5-class classification of different methods.

Method	Total Accuracy
1	0.6641
2	0.6934
3	0.6634
4	**0.6967**

## Data Availability

The data presented in this study are openly available in [ICBHI 2017] at [https://doi.org/10.1007/978-981-10-7419-6_6], reference number [33].

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
