# Peer review of "A New Shapley-Based Feature Selection Method in a Clinical Decision Support System for the Identification of Lung Diseases"

_diagnostics, 2023, doi:10.3390/diagnostics13233558_

Round 1
Reviewer 1 Report
Comments and Suggestions for Authors
This manuscript presents a compelling and timely study addressing the pressing issue of lung diseases, particularly in the aftermath of the Covid pandemic. The development of a clinical decision support system using the Decision Tree Algorithm (DTA) for the classification of various lung conditions is both innovative and pragmatic. The inclusion of five specific cases, including URTI, COPD, Pneumonia, and Bronchiolitis, adds granularity to the analysis.
The authors' choice of DTA as the classifier not only contributes to the accuracy of the system but also enhances the interpretability of the decision-making process. The incorporation of Shapley values for feature set selection is a noteworthy addition, reflecting a thoughtful approach to optimizing efficiency. The utilization of Power Spectral Density (PSD), Mel Frequency Cepstral Coefficients (MFCC), and statistical characteristics from lung sound recordings as feature sets underscores the comprehensive nature of the study.
Comments on the Quality of English LanguageThere is room for improvement in the quality of English language usage, particularly in the context of citing references. Ensuring consistency and precision in the citation format will enhance the overall readability and professionalism of the manuscript.
Reviewer 2 Report
Comments and Suggestions for Authors
The paper presents a lung disease diagnostic system utilizing a decision tree algorithm combined with Shapley value-based feature selection procedures. My comments are given below:
1. In the introduction, beginning with line 65, the author dedicates a paragraph to each referenced study, which seems unnecessary. Other than elaborating on the datasets, the accuracy, and other details of the other studies, the author should emphasize more on the differences and improvements of this work compared to the existing one. e.g., it is very difficult to tell if the new system indeed has better performance in lung disease classification after I read through the paper. Though the authors claim it to be one of the two goals.
2. I don't see advantages to having an interpretable system. From my reading, it seems like the accuracy (Table 5) is not as good as the ones mentioned in the introduction. From Section 7, it seems like the interpretability of the new system does not reveal any new mechanisms or have solid biological justifications. Given this and the potentially worse accuracy, I wonder the meaning of using this system.
3. The advantages of the Shaply value-based procedure are not clear; in addition to comparing the strategy of using all the raw features, some other more common feature selection strategies, from simple correlation-based strategies to more advanced regularization-based methods, could be compared.
4. Could the authors clarify whether feature selection was conducted as part of the cross-validation pipeline on the training data alone, or if it was performed prior to the cross-validation using the entire dataset. Implementing feature selection before cross-validation could result in data leakage, where information from the test set influences the training process, potentially leading to overly optimistic performance estimates.
5. The title of Section 7 is misleading, the author should consider using "results" or other titles instead.
Round 2
Reviewer 2 Report
Comments and Suggestions for Authors
I don't have further concerns.